# RNA Biomarkers as a Response Measure for Survival in Patients with Metastatic Castration-Resistant Prostate Cancer

**DOI:** 10.3390/cancers13246279

**Published:** 2021-12-14

**Authors:** Emmy Boerrigter, Guillemette E. Benoist, Inge M. van Oort, Gerald W. Verhaegh, Anton F. J. de Haan, Onno van Hooij, Levi Groen, Frank Smit, Irma M. Oving, Pieter de Mol, Tineke J. Smilde, Diederik M. Somford, Paul Hamberg, Vincent O. Dezentjé, Niven Mehra, Nielka P. van Erp, Jack A. Schalken

**Affiliations:** 1Radboud University Medical Center, Department of Pharmacy, Radboud Institute for Health Sciences, 6525 GA Nijmegen, The Netherlands; emmy.boerrigter@radboudumc.nl (E.B.); m.benoist@dz.nl (G.E.B.); 2Radboud University Medical Center, Department of Urology, Radboud Institute for Molecular Life Sciences, 6525 GA Nijmegen, The Netherlands; inge.vanoort@radboudumc.nl (I.M.v.O.); gerald.verhaegh@radboudumc.nl (G.W.V.); onno.vanhooij@radboudumc.nl (O.v.H.); levi.groen@radboudumc.nl (L.G.); jack.schalken@radboudumc.nl (J.A.S.); 3Radboud University Medical Center, Department for Health Evidence, Biostatistics, 6525 GA Nijmegen, The Netherlands; ton.dehaan@radboudumc.nl; 4MDxHealth, 6534 AT Nijmegen, The Netherlands; frank.smit@mdxhealth.com; 5Department of Medical Oncology, Ziekenhuisgroep Twente, 7609 PP Almelo, The Netherlands; i.oving@zgt.nl; 6Department of Medical Oncology, Gelderse Vallei Hospital, 6716 RP Ede, The Netherlands; molp@zgv.nl; 7Department of Medical Oncology, Jeroen Bosch Hospital, 5223 GZ ’s-Hertogenbosch, The Netherlands; t.smilde@jbz.nl; 8Department of Urology, Canisius Wilhelmina Hospital, 6532 SZ Nijmegen, The Netherlands; r.somford@cwz.nl; 9Department of Medical Oncology, Franciscus Gasthuis & Vlietland, 3045 PM Rotterdam, The Netherlands; P.Hamberg@Franciscus.nl; 10Department of Medical Oncology, Netherlands Cancer Institute, Antoni van Leeuwenhoek Hospital, 1066 CX Amsterdam, The Netherlands; v.dezentje@nki.nl; 11Radboud University Medical Center, Department of Medical Oncology, 6525 GA Nijmegen, The Netherlands; Niven.Mehra@radboudumc.nl

**Keywords:** castration-resistant prostate cancer, biomarkers, RNA, survival, abiraterone acetate, enzalutamide, liquid biopsy

## Abstract

**Simple Summary:**

Despite the increasing number of treatments for advanced prostate cancer, the evaluation of the added value of each line of treatment on survival outcome is challenging. Therefore, biomarkers to discriminate short-term survivors from long-term survivors shortly after start of treatment are urgently needed. We demonstrate that in 93 patients with mCRPC treated with first-line abiraterone acetate or enzalutamide the addition of *KLK3* and miR-375 (at baseline and 1 month) to standard clinical parameters resulted in the best prediction model for survival assessment.

**Abstract:**

Treatment evaluation in metastatic castration-resistant prostate cancer is challenging. There is an urgent need for biomarkers to discriminate short-term survivors from long-term survivors, shortly after treatment initiation. Thereto, the added value of early RNA biomarkers on predicting progression-free survival (PFS) and overall survival (OS) were explored. The RNA biomarkers: *KLK3* mRNA, miR-375, miR-3687, and *NAALADL2-AS2* were measured in 93 patients with mCRPC, before and 1 month after start of first-line abiraterone acetate or enzalutamide treatment, in two prospective clinical trials. The added value of the biomarkers to standard clinical parameters in predicting PFS and OS was tested by Harell’s C-index. To test whether the biomarkers were independent markers of PFS and OS, multivariate Cox regression was used. The best prediction model for PFS and OS was formed by adding miR-375 and *KLK3* (at baseline and 1 month) to standard clinical parameters. Baseline miR-375 and detectable *KLK3* after 1 month of therapy were independently related to shorter PFS, which was not observed for OS. In conclusion, the addition of *KLK3* and miR-375 (at baseline and 1 month) to standard clinical parameters resulted in the best prediction model for survival assessment.

## 1. Introduction

Metastatic castration-resistant prostate cancer (mCRPC) is characterized by disease progression despite suppression of gonadal androgen synthesis. Since maintained androgen receptor-mediated signaling is uncovered as a principal mechanism of mCRPC progression, novel androgen receptor signaling inhibitors (ARSI’s) were developed [1]. Enzalutamide and abiraterone acetate belong to the ARSI’s and are now widely used. Enzalutamide is a potent androgen receptor (AR) antagonist that competitively inhibits the binding of androgens to the AR, thereby blocking AR signaling and subsequently the proliferation of prostate cancer cells [2]. Abiraterone acetate is an inhibitor of cytochrome P450 17α-hydroxylase/17,20-lyase (CYP17A1), an enzyme essential in the process of androgen synthesis [3]. CYP17A1 inhibition prevents the formation of androgens in the adrenal glands, testis and prostate cancer cells. Both drugs have shown a comparable efficacy with an improved overall survival (OS) of 3 years in chemotherapy-naïve patients [4,5]. However, not all mCRPC patients benefit from these drugs, as approximately 25% of all patients starting first-line ARSI treatment do not show any response to these agents (i.e., intrinsic resistance). Eventually all responding patients will acquire resistance to ARSI’s [6,7].

Evaluation of treatment response in mCRPC patients remains challenging. According to the Prostate Cancer Working Group 3 (PCWG3) criteria, the first response evaluation should take place after 12 weeks of therapy by biochemical (i.e., serum prostate-specific antigen (PSA) changes), radiologic and clinical response assessment [8]. If progression is observed at the first evaluation moment, the patient is probably intrinsically resistant; particularly patients that deteriorate during these first months are those who are exposed to an ineffective line of treatment, with the potential risk of drug-related toxicity and financial burden. Therefore, it is essential to discriminate patients with short-term survival from long-term survival early after treatment initiation.

PFS is frequently used as primary endpoint to evaluate treatment effect, while the main and clinically most meaningful endpoint remains OS [9]. Particular in early setting of treatment for metastatic prostate cancer patients, PFS is used as a surrogate endpoint for OS, since it is earlier achieved and requires smaller groups of patients. However, in the quickly evolving prostate cancer treatment landscape many new drugs are available which disturbs the relation between PFS and OS outcomes. PSA changes are frequently used as an indicator of treatment response and surrogate marker for OS outcomes [8]. However, also this relation is disturbed due to introduction of many new treatment lines in prostate cancer. Therefore, neither PFS, nor PSA decline are strong indicators for OS in first-line patients. If we are able to find an early response measure for predicting OS benefit, clinicians can be guided to continue or early stop treatment. Currently, there is an unmet need for response indicators to predict OS benefit shortly after treatment initiation.

To discriminate between long-term survivors and short-term survivors, several prognostic models have been developed [10,11,12,13]. These prognostic models consist of different clinical parameters such as PSA, albumin, lactate dehydrogenase (LDH), hemoglobin (Hb) and alkaline phosphatase (ALP). However, the majority of models only include baseline parameters, while biomarker-dynamics after start of treatment are more likely to discriminate between patients who will benefit from treatment. Although promising, to date, no prognostic model is universally accepted for use in clinical practice, and better prognostic models are still needed to make treatment decisions.

Previous studies have shown that biomarkers can play an important role in identifying non-responders from long-term responders. For example, baseline detection of *androgen-receptor splice variant-7* (*AR-v7*) is a strong predictive marker for resistance to ARSI’s [14,15]. However, still some *AR-v7*-positive patients remain responsive to ARSI’s [16]. Therefore, risk assessment shortly following initiation of treatment may be needed to more robustly guide early decision making by identifying patients who are intrinsically resistant and won’t benefit from treatment. Circulating tumor cell (CTC) enumeration can be used to monitor treatment response [17]. However, due to the difficult sampling procedure and technical challenges, it is hard to implement CTC enumeration in clinical care of patients with mCRPC. Recently, studies investigating the potential use of circulating tumor DNA (ctDNA) as a predictive biomarker are in the spotlight [7,18]. Pre-therapy plasma ctDNA levels and changes after treatment initiation appear prognostic [19,20,21]. However, the robustness of ctDNA as an early marker for treatment response in prostate cancer patients still needs to be assessed.

In our two multi-center prospective clinical trials (ClinicalTrials.gov Identifier: NCT02426333 and ClinicalTrials.gov Identifier: NCT02471469) in which patients with mCRPC started first-line treatment with enzalutamide and abiraterone acetate respectively, several liquid biomarkers were measured longitudinally [22,23]. Four biomarkers were identified that were upregulated in mCRPC patients compared to healthy controls, the prostate-specific *kallikrein-related peptidase 3* (*KLK3*) mRNA, which codes for prostate-specific antigen (PSA), the microRNAs (miR) miR-375 and miR-3687, and the long non-coding RNA (lncRNA) *N-acetylated alpha-linked acidic dipeptidase like 2 antisense RNA 2* (*NAALADL2-AS2)* [22,23]. These biomarkers are relatively easy and inexpensive to quantify in blood. To discriminate between short-term survivors and long-term survivors, we explore the added value of these RNA biomarkers before and shortly after start of treatment to standard clinical parameters in predicting survival.

## 2. Materials and Methods

### 2.1. Study Design

Two prospective, observational, multicenter studies were conducted in several hospitals in the Netherlands (ClinicalTrials.gov: NCT02426333 and ILUMINATE: NCT02471469) [22,23]. Eligible patients were patients with mCRPC starting first-line enzalutamide or abiraterone acetate treatment in the ILUMINATE or OPTIMUM study respectively, according to the drug label. Pretreatment with upfront docetaxel according to CHAARTED/STAMPEDE protocols in a hormone-sensitive prostate cancer setting was allowed. The exclusion criteria for both studies were: co-medication that effected enzalutamide or abiraterone pharmacokinetics (e.g., potent CYP2C8 and CYP3A4 inhibitors and inducers), life expectance less than six months and non-measurable disease per PCWG3 criteria. If patients stopped treatment before the second visit (1 month after start of treatment) the patients were replaced. Both studies were conducted in accordance with Good Clinical Practice and the Declaration of Helsinki and approved by our Investigational Review board. Written informed consent was obtained from all patients before entering the study.

Thirty healthy individuals (10 men <35 year, 10 men between 55–70 year, and 10 women (no age restriction)), donated an aliquot of their blood, after written informed consent was given. Their blood was used as control, to correct for background levels of the RNAs [22,23].

### 2.2. Assessments

Patients had to visit the hospital before start and at 1 month, 3 months and 6 months after start of treatment for blood sampling and their routine care. Blood was collected in PAXgene Blood RNA Tubes (PreAnalytiX; Qiagen/BD-company, Hombrechtikon, Switzerland) for biomarker analysis at each visit. During the first 6 months of study, patients filled out a diary to check for drug adherence and side effects. After 6 months, patients were followed-up for PFS and OS data. To assess radiological progression, imaging was performed by computer tomography or magnetic resonance imaging and bone scans before start of treatment and at 3 and 6 months after start of therapy. Radiological response and progression were assessed by an independent central reviewer during the study period of 6 months according to Response Evaluation Criteria in Solid Tumors 1.1 criteria [24]. After the first 6 months of treatment, progression was assessed by the treating physician. Progression could be either radiographic, biochemical or clinically. Overall survival was defined as the time from start of treatment to death from any cause.

### 2.3. Biomarker Analysis and Selection

Biomarker analysis in whole blood RNA (PAXGENE tubes) for the ILUMINATE and OPTIMUM study, are described in detail by Benoist et al. and Boerrigter et al. respectively [22,23]. Based on the previous analysis of the biomarkers in the ILUMINATE and OPTIMUM study, only the biomarkers that were upregulated compared to healthy controls were selected for further analysis. This resulted in the selection of the following biomarkers: *KLK3*, miR-375 and miR-3687 and *NAALADL2-AS2*. *KLK3* refers to *KLK3* mRNA and PSA to serum PSA.

Since our objective was to identify early biomarkers for survival (PFS and OS), only the baseline values and the change (delta) in expression at 1 month was used for further analysis. The delta was calculated as (log relative expression at 1 month)−(log relative expression at baseline). The relative expression is defined as the expression in patients divided by the expression in healthy controls.

### 2.4. Statistical Analysis

To investigate the selected biomarkers as indicators for survival (PFS and OS), Kaplan-Meier analyses were used and survival time differences were compared with the use of log-rank tests. Furthermore, to explore whether biomarker dynamics can predict PFS and OS, the analysis were repeated for the delta values of the biomarkers. The cutoff values used for the biomarker expression at baseline and the delta values, were calculated by Maximally Selected Rank Statistics [25]. Since *KLK3* was only detectable in a limited number of patients at baseline and at 1 month, the cutoff was set at detectable yes/no at baseline and detectable yes/no at 1 month.

To explore whether the effect of biomarkers on survival was similar for abiraterone acetate and enzalutamide, Kaplan-Meier analyses for the biomarkers were repeated and compared for both treatments.

Next, a prognostic model for PFS and OS was built consisting of clinical parameters and the selected biomarkers. The clinical parameters used for the prognostic model were PSA, Hb, albumin, ALP and LDH. These parameters have previously been shown to be prognostic for survival and are used in prediction models for patients with mCRPC [10,11,12,13]. Furthermore, the PSA delta at 1 month was added as a clinical parameter to the model, since earlier research showed that PSA decline at 4 weeks is associated with OS [26]. All parameters were log-normalised. Owing to the small sample size and the limited number of events, we used the Least Absolute Shrinkage and Selection Operator (LASSO) method to select the most relevant clinical parameters [27]. Only the parameters that did not shrink to zero were used in the final clinical prognostic model.

Several prognostic models were build, starting first with a model consisting of only clinical parameters. Thereafter, the four selected RNA biomarkers (at baseline and 1 month) were added to the clinical model one by one. The predictive performance of each model was calculated by the concordance index (Harrell’s C-index) [28]. The concordance index ranges between 0.5 and 1.0, where a value of 1.0 means that the model can perfectly determine which patient will live longer, and a value of 0.5 means that the model is not able to discriminate between short and long-term survivors. If the predictive performance of the clinical model increased by addition of a biomarker, the concordance index of a combination of the biomarkers was tested. Finally, the parameters included in the model with the highest concordance index were selected for multivariate Cox regression to assess the effect of the different parameters on the prediction of PFS and OS.

Missing variables were imputed by multiple imputation using MICE package in R software (version 3.13.0) [29]. The amount of missing data was less than 5%. Data appeared to be missing at random, due to a failure in sample collection. Five imputed datasets were created using the variables: PSA, Hb, albumin, ALP, LDH, *KLK3* baseline, *KLK3* 1 month, miR-375 baseline, miR-375 delta, miR-3687 baseline, miR-3687 delta, *NAALADL2-AS2* baseline, *NAALADL2-AS2* delta and the Nelson-Aalen estimator [30,31]. To test the influence of imputation, the analysis was performed with and without imputed data. Results were comparable. All tests were two-sided, and *p* values of 0.05 or less were considered statistically significant. Since this was an exploratory study, no correction for multiple testing was done. Statistical analysis was performed with the use of R software, version 3.6.2.

## 3. Results

### 3.1. Patients

In total 93 patients were included in the studies, of which 40 were treated with enzalutamide and 53 were treated with abiraterone acetate. Patient characteristics are shown in Table 1. At time of analysis, after a median follow up of 27.4 months (range 3.0 to 62.4), 67% of the patients had died (*n* = 62) and 84% of the patients showed progression (*n* = 78), 7 patients stopped treatment due to toxicity or other reasons after the second visit at 1 month (censored for PFS and included in OS analysis) and 8 patients were still on treatment. The median time to progression was 14.5 months (95% confidence interval (CI); 11.8–17.2) and the median overall survival was 28.4 months (95% CI; 24.6–32.2).

### 3.2. Biomarkers as Indicator for Survival

#### 3.2.1. Baseline Biomarkers as Indicator for Progression Free Survival

Detectable levels of *KLK3* at baseline were related to shorter PFS compared to non-detectable *KLK3* levels, median time to PFS 8.0 months (95% CI; 5.8–12.3) vs. 16.4 months (95% CI; 14.8–22.6) (*p* = 0.00042) (Figure 1A). Patients with high levels of miR-375 at baseline also showed shorter PFS compared to patients with low levels of miR-375, median time to PFS 8.0 months (95% CI; 5.8–13.3) vs. 20.0 months (95% CI; 15.0–24.5) (*p* < 0.0001) (Figure 1B). A similar difference in PFS (including 90% CI) was seen between the patients with high vs. low expression of miR-3687 (Figure 1C). For *NAALADL2-AS2* no difference in median time to PFS was observed for patients with a low versus high expression at baseline a low expression at baseline, median time to PFS 14.5 months (95% CI; 11.5–17.9) vs. 13.4 (95% CI; 8.0–NA) (*p* = 0.077) (Figure 1D).

#### 3.2.2. Biomarker Dynamics at 1 Month as Indicator for Progression Free Survival

For *KLK3*, 17 patients with detectable levels of *KLK3* at baseline had undetectable levels at 1 month, only 1 patient had undetectable levels at baseline and detectable levels at 1 month. Detectable vs. undetectable levels of *KLK3* after 1 month of therapy was related to a median PFS of 4.6 months (95% CI; 2.72–NA) vs. 15.2 months (95% CI; 13.3–20.0) (*p* < 0.0001) (Figure 2A). For miR-375, delta values below the cutoff at 1 month were related to shorter PFS compared to delta values above the cut-off: 8.5 months (95% CI; 5.8–15.7) vs. 15.9 months (95% CI; 13.3–22.6) (*p* = 0.00088) (Figure 2B). For miR-3687, delta values below the cutoff were related to shorter PFS compared to delta values above the cutoff after 1 month of therapy: 13.3 months (95% CI; 11.0–15.7) vs. 22.9 months (95% CI; 17.9–NA) (*p* = 0.019) (Figure 2C). For *NAALADL2-AS2* no effect of 1 month dynamics in biomarkers (increase or decrease) was observed (*p* = 0.1) (Figure 2D).

#### 3.2.3. Baseline Biomarkers as Indicator for Overall Survival

Detectable levels of *KLK3* at baseline were related to shorter OS compared to non-detectable *KLK3* levels, median time to OS 20.5 months (95% CI; 15.4–30.3) vs. 31.7 months (95% CI; 27.3–44.9) (*p* = 0.005) (Figure 3A). Patients with high levels of miR-375 at baseline also showed shorter OS compared to patients with low miR-375 levels, median time to OS 23.0 months (95% CI; 19.8–27.6) vs. 40.5 months (95% CI; 30.3–NA) (*p* < 0.001) (Figure 3B). The same relation was seen for the expression of miR-3687; high levels of miR-3687 were related to shorter OS compared to patients with low levels, median time to OS 23.5 months (95% CI; 17.7–28.6) compared to 39.0 months (95% CI; 28.4–NA) (*p* < 0.001) (Figure 3C). For *NAALADL2-AS2* a low expression at baseline showed a trend for shorter OS compared to a high expression, median time to OS 21.9 months (95% CI; 18.8–44.9) vs. 30.7 (95% CI; 27.6–43.4) (*p* = 0.088) (Figure 3D).

#### 3.2.4. Biomarker Dynamics at 1 Month as Indicator for Overall Survival

Detectable vs. undetectable levels of *KLK3* after 1 month of therapy was related to a median OS of 15.9 months (95% CI; 8.0–NA) vs. 30.3 months (95% CI; 26.8–43.4) (*p* = 0.0016) (Figure 4A). For the three other biomarkers delta values below the cutoff at 1 month were related to shorter OS compared to delta values above the cutoff: miR-375 22.9 months (95% CI; 16.0–27.6) vs. 32.9 months (95% CI; 26.3–NA) (*p* < 0.001) (Figure 4B); miR-3687 23.1 months (95% CI; 19.8–30.7) vs. 32.2 months (95% CI; 26.3–NA) (*p* = 0.0076) (Figure 4C); *NAALADL2-AS2* 22.1 months (95% CI; 15.9–NA) vs. 29.9 months (95% CI; 26.3–44.9) (*p* = 0.016), respectively (Figure 4D).

The cutoff values for the Kaplan-Meier analyses were set at detectable (yes/no) for KLK3. For the biomarkers miR-375, miR-3687 and *NAALADL2-AS2* that are also detectable in healthy volunteers (Appendix A) the cut-off values were calculated relative to healthy controls by using maximally selected rank statistics (Appendix A). The biomarker expression for abiraterone acetate versus enzalutamide treated patients was comparable (Appendix A).

### 3.3. Predictive Performance of Different Models

The predictive performance of the clinical model consisting of baseline PSA, Hb, ALP, LDH and delta PSA after 1 month of therapy, showed a concordance of 0.71 and 0.70 for PFS and OS respectively. Lasso penalization revealed that albumin had no effect on the predicting performance and was therefore deleted from the clinical model. When the biomarkers *KLK3*, miR-375 and *NAALADL2-AS2* (at baseline and 1 month) were added, there was an increase in concordance of the model to 0.73, 0.74 and 0.73, respectively for PFS, and to 0.72, 0.72 and 0.71, respectively for OS. Addition of miR-3687 (at baseline and 1 month) increased the concordance for PFS to 0.73, but did not show any increase in the predictive performance of the model for OS. Finally, the combination of biomarkers was tested. Combination of miR-375 and *KLK3* (both at baseline and 1 month) showed the highest concordance of 0.74 for PFS and OS. Addition of *NAALADL2-AS2* did not increase the concordance of this model any further. Therefore, the best predictive model consisted of miR-375 and *KLK3* (both at baseline and 1 month) together with the clinical parameters (Table 2, model 6).

### 3.4. Multivariate Cox Regression of the Predictive Model

To determine whether these biomarkers were independently prognostic markers for PFS and OS, a multivariate Cox regression analyses was performed. This analysis revealed that baseline Hb, delta PSA at 1 month and the biomarkers KLK3 at 1 month and baseline miR-375 were independent markers for PFS. For OS, only baseline LDH and baseline Hb were independent prognostic markers. Table 3 shows the hazard ratio’s and 95% CI of the selected covariates. 

## 4. Discussion

In this study, we investigated the added value of RNA biomarkers measured in liquid biopsies before start and after 1 month of treatment, in patients with mCRPC treated with first-line abiraterone acetate or enzalutamide, in predicting survival. Early identification of patients with short-term treatment benefit in terms of PFS and OS, is needed to change the treatment strategy of these patients within a favorable time window. Therefore, markers that predict short lasting benefit early after start of treatment are essential. Our study revealed that the addition of two biomarkers, *KLK3* mRNA and miR-375, measured before start and 1 month after start of treatment, can improve the prediction of survival.

This is the first study exploring the added value of RNA biomarker dynamics in assessing survival shortly after start of treatment. In earlier prognostic models mostly baseline parameters were considered [10,11,12,13]. However, by only incorporating baseline markers no distinction between short-term survivors and long-term survivors after treatment initiation can be made. Predictive biomarkers (e.g., *AR-v7*, AR copy number gain) might be able to select intrinsically resistant patients [14,15,18,32]. However, still some patients with detectable *AR-v7* or AR copy number gain do respond to ARSI treatment. Biomarker dynamics shortly after start of therapy are urgently needed to evaluate treatment benefit. Therefore, CTC dynamics after start of treatment has gained interest over the past years. CTC conversion at 13 weeks seems to be a good marker for treatment response [33]. Furthermore, a 30% CTC decline already after 4 weeks of treatment was associated with prolonged OS [34]. However, since CTC enumeration is challenging to implement in clinical practice, there is a need for biomarkers that are feasible to implement in routine outpatient care.

The median OS rates found in our study are slightly shorter compared to the reported OS rates from the registration studies of enzalutamide and abiraterone acetate (26.8 vs. 35.5 months and 28.4 vs. 34.7 months respectively) [35,36]. Though, our median survival rates are in line with a large previously published real-life cohort of chemotherapy-naïve patients treated with enzalutamide or abiraterone acetate (median OS 29.6 and 25.9 months respectively) [37]. Our cohort is therefore considered representative for a real-life first line treated cohort of patients with mCRPC.

Biomarkers that can be measured short after start of therapy have the potential to early identify patients with short-term treatment benefit, before regular clinical assessment after approximately 12 weeks of therapy. Therefore, in our model we have included RNA biomarkers measured 1 month after treatment initiation. Earlier studies have identified common prognostic clinical parameters [10,11,12,13]. Therefore, in our study we have added the biomarkers to these earlier identified clinical parameters. Our final prediction model consisted of the following parameters: PSA, Hb, ALP, LDH, delta PSA at 1 month, *KLK3* detection at baseline, *KLK3* detection after 1 month of therapy, miR-375 levels at baseline and delta miR-375 after 1 month of therapy. A previous study revealed alternative biomarkers to assess survival in mCRPC patients, consisting of comparable clinical parameters: PSA, Hb, ALP, LDH, albumin and PSA change at 13 weeks, with the addition of CTC at baseline and CTC change after 13 weeks of therapy [38]. The concordance (C-index) of their model was in line with our model (0.75 versus 0.74, respectively). Although our model is marginally less predictive it makes early treatment decision possible already 1 month after start of therapy. Furthermore, CTC monitoring is challenging due to the difficult sampling procedure and the low abundance of CTC’s in early stages of diseases. Our biomarkers are detectable in earlier stages of disease and liquid RNA biopsies are relatively easy to collect and store, allowing for broad implementation in clinical practice. However, our model still needs to be improved to be able to more accurately discriminate between short-term vs. long-term survivors. Since the ctDNA quantity seems a promising marker for treatment response, possibly the addition of ctDNA to our model could significantly improve the predictive performance. Further research is needed to investigate what the effect is of adding ctDNA to our model.

Univariate analysis showed that baseline and delta values at 1 month of *KLK3*, miR-375 and miR-3687 were the most promising markers to further investigate in our predictive model. The addition of *KLK3* and miR-375 (both at baseline and 1 month) resulted in the most accurate model to predict PFS and OS. *KLK3* detection at 1 month and miR-375 expression at baseline were independent prognostic markers for PFS, however both biomarkers were not identified as independent markers for OS after correcting for clinical parameters. This apparent discrepancy can potentially be explained by the fact that these biomarkers are predictive for response to ARSI’s but not related to response to other lines of treatment. This hypothesis is opposed by another study, which investigated the relation between circulating miR-375 and treatment outcome in mCRPC patients treated with abiraterone or docetaxel. This study identified that high baseline levels of miR-375 were significantly related to shorter OS in the docetaxel cohort and to radiologic progression in the abiraterone cohort [39]. Further research is needed to investigate whether these biomarkers are predictive markers for ARSI’s or just have prognostic value.

*KLK3* is the mRNA transcript of PSA. Since progression was defined as either biochemical, clinical or radiographic, *KLK3* detection could be a surrogate of biochemical progression. Still, after correcting for PSA decline at 1 month, *KLK3* presence at 1 month was an independent marker of PFS. This might be explained by the hypothesis that *KLK3* reflects CTC’s, since most of the CTC’s are *KLK3* positive [23] while PSA is most likely secreted by tumor masses in the body. However, this hypothesis still needs to be confirmed.

The functionality of miR-375 is not fully elucidated yet. It is shown that miR-375 is upregulated in various types of cancer [40]. MiR-375 can be used as biomarker to distinguish between prostate cancer and healthy individuals or benign prostatic hyperplasia [41,42,43]. Furthermore, it was shown that miR-375 is significantly upregulated in a more aggressive state of prostate cancer and might be related to the development of metastasis [44,45]. Thus, it is thought that miR-375 is a marker for tumor load. In line with these findings and the study by Zedan et al., we show that upregulation of miR-375 at baseline is related to shorter survival [39]. Therefore, we hypothesized that a larger decrease after start of treatment might be an indicator of treatment response and related to longer survival. However, we couldn’t confirm this hypothesis since our data showed that a large decrease (more than 35%) in miR-375 after 1 month of treatment was a marker for worse survival. Elucidating the functionality of miR-375 is needed to explain this yet unclarified observation.

Our biomarkers were only measured in patients starting ARSI treatment. An independent cohort of patients treated with a different line of treatment to compare the predictive performance of the identified biomarkers is warranted. However, our approach of using easy to measure biomarkers after 1 month of therapy for monitoring treatment response was shown to be feasible and this approach is promising in selecting patients that will benefit most from therapy.

## 5. Conclusions

In conclusion, this is the first study using early RNA biomarker dynamics in addition to clinical parameters to predict PFS and OS in first-line patients treated with ARSI’s. The addition of *KLK3* and miR-375 (both at baseline and 1 month) to standard clinical parameters resulted in a better model to assess PFS and OS after treatment initiation. The treatment landscape for prostate cancer is rapidly changing and many new treatment options are available. To minimize patient exposure to less effective treatment and improve treatment outcome in terms of survival benefit and quality of life, it is important to be able to timely switch or stop treatment in case of intrinsic resistance. Therefore, early biomarkers are needed to predict survival benefit shortly after treatment initiation.

## Figures and Tables

**Figure 1 cancers-13-06279-f001:**
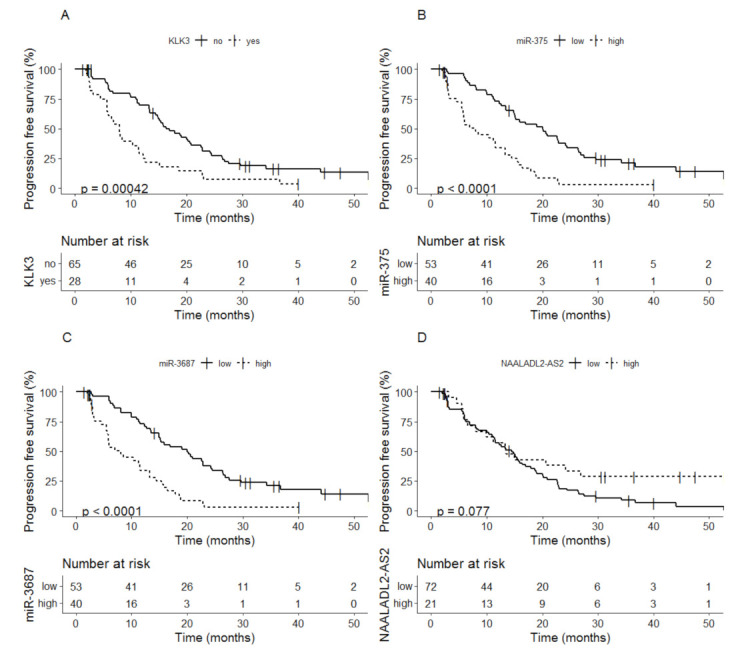
Kaplan-Meier analysis of progression-free survival based on baseline values of (**A**) *KLK3*, (**B**) miR-375, (**C**) miR-3687 and (**D**) *NAALADL2-AS2*.

**Figure 2 cancers-13-06279-f002:**
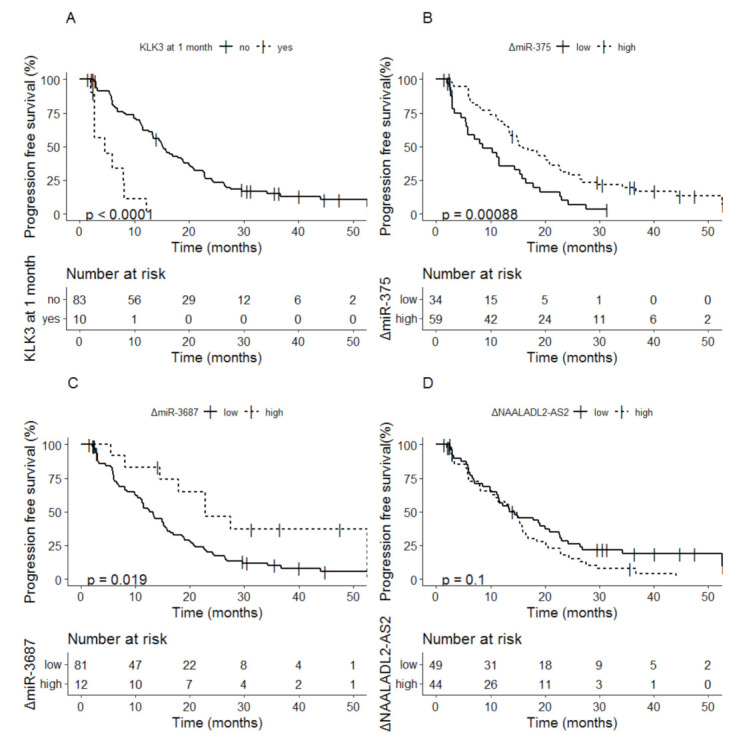
Kaplan-Meier analysis of progression-free survival based on delta values at 1 month of (**A**). *KLK3*, (**B**) miR-375, (**C**) miR-3687 and (**D**) *NAALADL2-AS2*.

**Figure 3 cancers-13-06279-f003:**
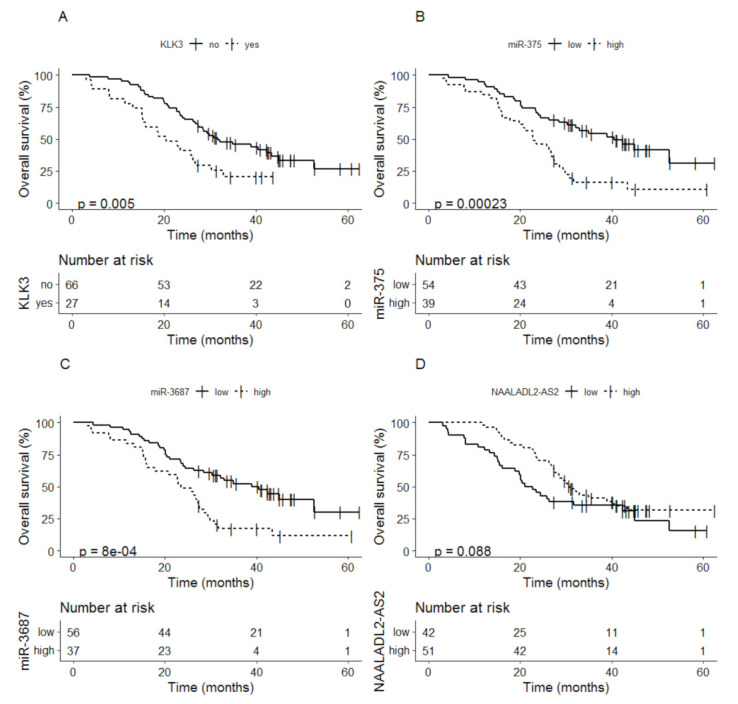
Kaplan-Meier analysis of overall survival based on baseline values of (**A**) *KLK3*, (**B**) miR-375, (**C**) miR-3687 and (**D**) *NAALADL2-AS2*.

**Figure 4 cancers-13-06279-f004:**
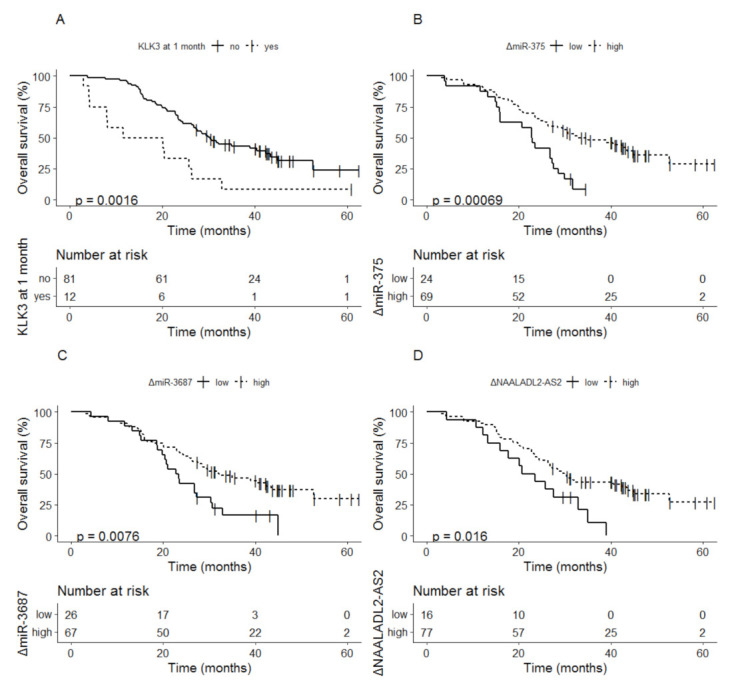
Kaplan-Meier analysis of overall survival based on delta values at 1 month for (**A**) *KLK3*, (**B**) miR-375, (**C**) miR-3687, and (**D**) *NAALADL2-AS2*.

**Table 1 cancers-13-06279-t001:** Baseline characteristics.

Patient Characteristics at Baseline	Total (*n* = 93)	Iluminate Enzalutamide (*n* = 40)	Optimum Abiraterone Acetate (*n* = 53)
Age at baseline (years)	73 (67–78)	74 (69–78)	71 (65–78)
Weight at baseline (kg)	85.7 (79.0–92.5)	85 (78–91)	86 (80–93)
Hb (mmol/L)	8.0 (7.5–8.5)	8.1 (7.6–8.5)	7.9 (7.4–8.4)
LDH (U/L)	230.0 (201–264)	230 (193–261)	230 (206–264)
ALP (U/L)	91.0 (73–125)	94 (76–125)	85 (71–125)
Albumin (g/L)	41 (37–43)	41 (37–43)	41 (37–44)
PSA (ng/mL)	45.0 (23.0–110)	49 (23–95)	39 (23–130)
PSA doubling time (months)	3.3 (2.2–5.8)	4.1 (2.3–6.4)	3.3 (2.3–6.1)
DHEAS (µmol/L)	1.7 (1.0–3.1)	1.9 (1.1–3.6)	1.6 (0.9–2.5)
Gleason score at diagnosis			
≤7	31 (33.3)	19 (47.5)	13 (24.5)
≥8	58 (62.4)	20 (50)	37 (69.8)
Missing	4 (4.3)	1 (2.5)	3 (5.7)
Ethnicity/Race			
White	89 (95.7)	36 (90.0)	53 (100)
Asian	2 (2.2)	2 (5.0)	
Missing	2 (2.2)	2 (5.0)	
ECOG performance status			
0	62 (67.4)	28 (70.0)	34 (64.2)
1	27 (29.3)	10 (25.0)	17 (32.1)
2	3 (3.3)	1 (2.5)	2 (3.8)
Missing	1 (1.1)	1 (2.5)	
Pre-treatment docetaxel ^a^	19 (20.4)	3 (7.5)	16 (30.2)
Previous treatments			
Prostatectomy	41 (44.1)	21 (52.5)	20 (37.7)
Radiation	38 (40.9)	17 (42.5)	21 (39.6)
Anti-androgenpre-treatment	49 (52.7)	11 (27.5)	38 (71.7)
Other ^b^	4 (4.3)	3 (7.5)	1 (1.9)
Missing	34 (36.6)	19 (47.5)	15 (28.3)
Spread of disease			
Lymph only	14 (15.1)	9 (22.5)	5 (9.4)
Bone only	25 (26.9)	11 (27.5)	14 (26.4)
Both bone and lymph	36 (38.7)	12 (30.0)	24 (45.3)
Visceral + lymph node/bone	16 (17.2)	7 (17.5)	9 (17.0)

Median time to progressionMonths (95% CI)	14.5 (11.8–17.2)	15.2 (6.1–24.3)	13.4 (10.0–16.8)
Median overall survivalMonths (95% CI)	28.4 (24.6–32.2)	26.8 (18.3–35.2)	28.4 (23.9–33.0)
Median follow-up timeMonths (range)	27.4 (3.0–62.4)	27.7 (7.8–62.4)	27.4 (3.0–60.8)

Abbreviations: HB, hemoglobin; LDH, lactate dehydrogenase; ALP, alkaline phosphatase; PSA, prostate-specific antigen; DHEAS, dehydroepiandrosterone sulfate. Data are presented as median (Q1-Q3) for continuous data or *n* (%) for categorical data. ^a^: pre-treatment with docetaxel according to CHAARTED/STAMPEDE schedule, ^b^: Tamoxifen, radium-223, dendritic cell vaccination, diethylstilbestrol.

**Table 2 cancers-13-06279-t002:** Predictive performance of the different prediction models for progression free survival and overall survival.

Model Variables	Progression Free Survival		Overall Survival	
	Harell’s C-Index ^#^	Standard Error	Harell’s C-Index	Standard Error
Model 1: clinical parameters * only	0.71	0.030	0.70	0.036
Model 2: model 1 + *KLK3*	0.73	0.029	0.72	0.034
Model 3: model 1 + miR-375	0.74	0.029	0.72	0.035
Model 4: model 1 + miR-3687	0.73	0.031	0.70	0.035
Model 5: model 1 + *NAALADL2-AS2*	0.71	0.030	0.71	0.036
Model 6: model 1 + miR-375 + *KLK3*	0.74	0.028	0.74	0.034
Model 7: model 1 + miR-375 + *NAALADL2-AS2*	0.73	0.030	0.73	0.035
Model 8: model 1 + *KLK3* + *NAALADL2-AS2*	0.72	0.030	0.72	0.035
Model 9: model 1 + miR-375 + *KLK3* + *NAALADL2-AS2*	0.74	0.029	0.74	0.034

**^#^** Concordance as calculated using the Harell’s C-index; ***** Clinical parameters were log-normalised baseline values of prostate specific antigen (PSA), hemoglobin, alkaline phosphatase, lactate dehydrogenase and PSA delta at 1 month. For the biomarkers the log-normalised baseline values and delta’s at 1 month were added.

**Table 3 cancers-13-06279-t003:** Multivariate Cox regression analysis in relation to progression free survival and overall survival.

Variables	Progression Free Survival		Overall Survival	
	HR (95% CI)	*p*-Value	HR (95% CI)	*p-*Value
Baseline log PSA	1.29 (0.83–2.00)	0.26	1.32 (0.79–2.20)	0.28
Baseline log Hb	9.06 × 10^−5^ (7.84 × 10^−7^–0.01)	<0.001	5.81 × 10^−6^ (3.37 × 10^−8^–1.00 × 10^−3^)	<0.001
Baseline log ALP	1.71 (0.55–5.29)	0.35	2.46 (0.71–8.47)	0.15
Baseline log LDH	0.23 (1.12 × 10^−2^–4.56)	0.32	0.02 (9.17 × 10^−4^–0.34)	0.009
Delta PSA at 1 month	2.97 (1.65–5.34)	<0.001	1.65 (0.93–2.91)	0.08
*KLK3* detectable at baseline	0.89 (0.42–1.86)	0.75	1.53 (0.75–3.14)	0.24
*KLK3* detectable at 1 month	4.00 (1.47–10.85)	0.007	1.50 (0.58–3.90)	0.40
Baseline log miR-375	1.93 (1.26–2.95)	0.003	1.27 (0.78–2.08)	0.33
Delta miR-375 at 1 month	0.97 (0.65–1.46)	0.89	0.75 (0.44–1.28)	0.29

Abbreviations: PSA; prostate specific antigen, Hb; hemoglobin, ALP; alkaline phosphatase and LDH; lactate dehydrogenase.

## Data Availability

The data that support the findings of this study are available from the corresponding author upon reasonable request.

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
