# Peer review of "RNA Biomarkers as a Response Measure for Survival in Patients with Metastatic Castration-Resistant Prostate Cancer"

_cancers, 2021, doi:10.3390/cancers13246279_

Round 1
Reviewer 1 Report
The authors of this manuscript describe the use of RNA biomarkers in blood as early predictors of treatment response. Two biomarkers (KLK3 and miR375) when measured at baseline and month 3 show predictive value for PFS and OS. When added to a model using clinical parameters it resulted in improvement of the prediction model. The findings and scientific approaches of this manuscript are a nice addition to the emerging field of liquid biopsies RNA biomarkers. New biomarkers to predict treatment response early on are urgently needed and the two described markers are promising candidates for clinical implementation.
Author Response
We thank the reviewer for the comments and for underscoring the importance of our research.
Reviewer 2 Report
The current article "RNA biomarkers as a response measure for survival in patients 2 with metastatic castration-resistant prostate cancer", discuss about the possibility(ies) of using established basic science markers KLK3, miR-375, miR-3687 and NAALADL2-AS2 as a clinical tool to predict the outcome of treatment in mCRPC. Authors have shown that addition two RNA biomarkers, KLK3 and miR-375, as a baseline at the start and post 1 month treatment can predict the progression free survival (PFS) and overall survival (OS). Though the number of patients in the study are less, the study provides a good predictive model which can be optimized for future use and opens up new possibilities in the field. All together the experiments were planned in a good ethical way, however, there is need of some clarification of the following questions.
- In figure 1, and 2, authors talked about the baseline levels of KLK3, miR-375, miR-3687 and NAALADL2-AS2 and detection of KLK3 and and differential expression of miR-375, miR-3687 and NAALADL2-AS2 after one month of treatment and found that there is negative correlation of PFS with detection of KLK3 both at the start of and one month after the treatment, similar correlation were observed with less expression of miR-375 and 3687 at the start and patients with higher del values post one month treatment, however, NAALADL2-AS2 did not show any significant correlation. Similar correlation were also observed in case of OS. Are these observation absolute, or any exception was observed?
Author Response
Response 1.
We thank the reviewer for the comments and the question. We have calculated a cutoff value for the baseline biomarker expression levels and delta values at 1 month of miR-375, miR-3687 and NAALADL2-AS2 by maximally selected rank statistics. Since KLK3 was only detectable in a limited number of patients at baseline and at 1 month, the cutoff was set at detectable yes/no at baseline and detectable yes/no at 1 month. The calculated cutoff values are provided in the supplementary material. At baseline we have observed that levels above the cutoff value (or detectable KLK3) was related to shorter PFS and OS. At 1 month, we have observed that a large decrease in expression levels was related to shorter PFS and OS. This is not an absolute observation per individual patient, but a determined increased risk for shorter survival for the whole group of patients with a large decrease in biomarker level at 1 month. The median PFS and OS of the group of patients below the cutoff was shorter compared to the group of patients above the cutoff. This observation is shown by Kaplan-Meier analysis and the median PFS and OS with 95% CI (see section 3.2.2.).
Reviewer 3 Report
In the manuscript entitled “RNA biomarkers as a response measure for survival in patients with metastatic castration-resistant prostate cancer” by Boerrigter et al, the authors have mainly established the added value of RNA biomarkers to the existing standard clinical biomarker panel in castration resistant metastatic cancer for determination of disease progression as well as disease therapeutic response. The added value of the biomarkers to standard parameters were specifically tested in predicting the overall survival (OS) and progression free survival (PFS) among patients with metastatic castration resistant prostate cancer. Overall it is a very important work which has been meticulously designed and very well executed. However, there are some concerns that need to be addressed.
Figure 1, Kaplan-Meier curve describes the PFS based on the baseline value of 4 RNA biomarkers studied. It is not clear how the baseline values were determined and also what were the cutoff values for change in expression (Figure 2) of these markers. The cutoff values for miR-375, miR-3687 and NAALADL2-AS2 at baseline, the description provided in the supplementary part of this study describes these values are in relation to healthy controls; however no healthy control or inclusion of healthy controls was mentioned in the main text.
Table 1 describes the baseline characteristics for patient population selected for this study. In the patient categories there are number factors that might potentially influence the progression or overall survival among these patients for e.g. prior treatment of these patients with chemotherapeutic agents such as Docetaxel or pretreatment with anti-androgen. Specifically Docetaxel has been previously shown to alter the level of miR-375 in patients with metastatic prostate cancer. Was this taken into account? Similar concern remains for KLK3. If yes, how was this adjusted for these patients?
For determination of predictive performance for PFS and OS (Table 2) the description of models i.e combining these RNA markers with previously described clinical markers is not clear.
Author Response
Point 1: Figure 1, Kaplan-Meier curve describes the PFS based on the baseline value of 4 RNA biomarkers studied. It is not clear how the baseline values were determined and also what were the cutoff values for change in expression (Figure 2) of these markers. The cutoff values for miR-375, miR-3687 and NAALADL2-AS2 at baseline, the description provided in the supplementary part of this study describes these values are in relation to healthy controls; however no healthy control or inclusion of healthy controls was mentioned in the main text.
Response 1: We thank the reviewer for this question. As the reviewer states, the cutoff values are shown in the supplementary material of our manuscript and the expression levels of the biomarkers were calculated relative to healthy controls.
The absolute values measured in healthy controls are reported in the earlier analyses (Benoist et al Clin Chem 2020 Jun 1:66(6):842-851 and Boerrigter et al. Mol Oncol 2021 Sep; 15(9):2453-2465). We can add these absolute values from the other articles into our supplementary material, if this reviewer prefers.
Based upon the question of the reviewer, we have added some additional information to our main text to describe the inclusion of our healthy control population and refer to the above mentioned manuscripts.
See lines 133-135: Thirty healthy individuals (10 men <35 year, 10 men between 55-70 year, and 10 women [no age restriction]), donated an aliquot of their blood, after written informed consent was given. Their blood was used as control, to correct for background levels of the RNAs [23, 24].
Point 2: Table 1 describes the baseline characteristics for patient population selected for this study. In the patient categories there are number factors that might potentially influence the progression or overall survival among these patients for e.g. prior treatment of these patients with chemotherapeutic agents such as Docetaxel or pretreatment with anti-androgen. Specifically Docetaxel has been previously shown to alter the level of miR-375 in patients with metastatic prostate cancer. Was this taken into account? Similar concern remains for KLK3. If yes, how was this adjusted for these patients?
Response 2: We thank the reviewer for addressing this important question. Our population consists of only first-line mCRPC patients. All patients were considered castration-resistant when entering our studies. So all patients had anti-androgen pretreatment or orchidectomy leading to castration levels of testosterone which was an inclusion criteria for our CRPC population. However, the use of ADT in our study was not accurately reported leading a high percentage of missing data.
Furthermore, all included patients were not treated with docetaxel in mCRPC setting before starting abiraterone acetate or enzalutamide. There were 19 (out of 93) patients treated with upfront docetaxel (6 cycles) in mHSPC setting. As far as we know there is no evidence that these patients have higher levels of miR-375 or KLK3. Patients treated with upfront docetaxel + ADT compared to patients treated with ADT alone show similar efficacy. (ref. Clin Genitourin Cancer. 2018 Apr;16(2):130-134)
Point 3: For determination of predictive performance for PFS and OS (Table 2) the description of models i.e combining these RNA markers with previously described clinical markers is not clear.
Response 3: We thank the reviewer for this comment. The description of the models is described in section 2.4 Statistical analysis. We have now clarified how the predictive performance of each model is calculated.
See lines 185-191: Several prognostic models were build, starting first with a model consisting of only clinical parameters. Thereafter, the four selected RNA biomarkers (at baseline and 1 month) were added to the clinical model one by one. The predictive performance of each model was calculated by the concordance index (Harrell’s C-index)[29]. The concordance index ranges between 0.5 and 1.0, where a value of 1.0 means that the model can perfectly determine which patient will live longer, and a value of 0.5 means that the model is not able to discriminate between short and long-term survivors. If the predictive performance of the clinical model increased by addition of a biomarker, the concordance index of a combination of the biomarkers was tested. Finally, the parameters included in the model with the highest concordance index were selected for multivariate Cox regression to assess the effect of the different parameters on the prediction of PFS and OS.
Round 2
Reviewer 3 Report
The revised version of the manuscript is much improved now. But there are some concerns.
- References need to be checked. References provided do not match the context in several occasions.
- The authors need to add the absolute cutoff values from the other articles into their supplementary material.
Author Response
Point 1: References need to be checked. References provided do not match the context in several occasions.
Response 1: We thank the reviewer for addressing this important point. We have now checked all the references and adapted the reference order.
Point 2: The authors need to add the absolute cutoff values from the other articles into their supplementary material.
Response 2: We thank the reviewer for this question. We have added the absolute values of the several RNA biomarkers measured in healthy volunteers in our supplementary material.